# Checking Functional Modularity in DNN By Biclustering Task-specific Hidden Neurons

**Jialin Lu**     **Martin Ester**
School of Computing Science
Simon Fraser University

## 1 Introduction

While real brain networks exhibit functional modularity, we investigate whether functional modularity also exists in Deep Neural Networks (DNN) trained through back-propagation. Under the hypothesis that DNN are also organized in task-specific modules, in this paper we seek to dissect a hidden layer into disjoint groups of task-specific hidden neurons with the help of relatively well-studied neuron attribution methods. By saying task-specific, we mean the hidden neurons in the same group are functionally related for predicting a set of similar data samples, i.e. samples with similar feature patterns.

We argue that such groups of neurons which we call Functional Modules can serve as the basic functional unit in DNN. We propose a preliminary method to identify Functional Modules via biclustering attribution scores of hidden neurons.

We find that first, unsurprisingly, the functional neurons are highly sparse, i.e., only a small subset of neurons are important for predicting a small subset of data samples and, while we do not use any label supervision, samples corresponding to the same group (bicluster) show surprisingly coherent feature patterns. We also show that these Functional Modules perform a critical role in discriminating data samples through ablation experiment. Also, these modules learn rich representations and are able to detect certain feature patterns demonstrated in a visual classification example.

## 2 Related Works

Modularity is generally encountered across a broad range of networks, including real brain neuronal networks, which means that the entire population of neurons can be parcellated into internally dense and externally sparse groups called modules or communities. And since that, researchers naturally think artificial neural networks also exhibits modularity, for example, Hinton et al. [3] put forward *neuron co-adaption* that some hidden neurons in the same layer co-adapt together as a module for prediction. Co-adaption is only discussed in thought experiments supported by some biological inspirations and we do not know how to identify such co-adaptive neurons. However this phenomenon itself inspires Dropout [3; 7] which is arguably the most robust regularization technique for DNN.

Identifying modularity in DNN remains difficult as community detection methods are generally not directly applicable on densely-connected acyclic graphs. We turn to the relatively well-studied neuron attribution methods and biclustering algorithm for help.

**Biclustering.**   is a data mining technique that simultaneously clusters the rows and columns of a matrix, and is especially popular in bioinformatics. We favor biclustering instead the standard clustering methods, which only cluster the rows or the columns of a matrix, for two reasons: 1) Practically, standard clustering methods can easily fail because of the curse of dimensionality. 2) From the hypothesis we really do not expect two neurons in the same neuron group to be similar for every stimulus (input data sample). We only expect neurons in the same group are functionally related for predicting a subset of similar data samples. Hopefully, this subset of samples should share similar feature patterns. That being said, there are many biclustering algorithms for different purposes as it is a very active research field. We choose to use spectral co-clustering because it

33rd Conference on Neural Information Processing Systems (NeurIPS 2019), Vancouver, Canada.

produces biclusters of strong connections with no overlaps, which is the simplest case for us to start with.

**Attribution methods for hidden neurons.** Neuron attribution, assigning a importance score to a neuron, is easier to do for artificial neural networks than real neural nets. Basically, the goal of neuron attribution is to assign a score $a$ to a hidden neuron $n$ that represents how much important this neuron is for predicting a sample $x$ to a class $y$. In a sense, the attribution score measures the per-sample importance of a hidden neuron. While for real neurons this can be computed as, for example, Pearson correlation to some task, for DNN we can utilize the internal weights and the feedforward structure of DNN to compute attribution scores.

There is a variety of neuron attribution methods. To name a few, Shrikumar et al. [5] proposed DeepLIFT, originally designed to assign scores to input nodes and can be generalized to assign a score of importance to a specific hidden neuron. Sundararajan et al. [8] proposed integrated gradients for attributing input neurons and later generalized it to total conductance [6; 1] for attribution hidden neurons. Leino et al. [4] proposed an influence-based attribution method. Such methods have been demonstrated to be capable of identifying important hidden neurons that are relevant to a specific prediction on a class $y$ given a data sample $x$. However, these methods do not consider that hidden neurons may be functionally related.

## 3 Method

At a high-level, our approach for finding Functional Modules in hidden neurons goes in two phases: 1) we first construct a neuron-sample matrix where each entry is a attribution score of a hidden neuron (row) for a input sample (column) and then 2) based on this matrix, we apply spectral co-clustering [2], a biclustering algorithm to simultaneously group neurons and samples that have consistent high attribution values.

Given a dataset $X = \{x\}^N$ of data samples, a pre-trained DNN $f$ parameterized by $\theta$ such that $\hat{y} = f_\theta(x)$ is the prediction of sample $x$, and an attribution function $a$ which can be any of [1; 4; 5; 6], the attribution score $a(n, x, y)$ measures how much a hidden neuron $n$ contributes to the prediction of sample $x$ into class $y$.

**Construct the neuron-sample attribution matrix.** We first specify the neurons to be from a given layer $l$ and construct the neuron-sample matrix $M_{N_l * N_{data}}$ where $N = |l|$ is the number of neurons in that layer and $N_{data} = |X|$ be the size of dataset $X = x_1, x_2, \ldots, x_N$. Each entry in the matrix represents $e_{ij} = e(n_i, x_j)$, where $e$ is a embedding function measuring the contribution of neuron $n_i$ for predicting data sample $x_j$. We compute $e(n_i, x_j) = e_{ij} = a(n_i, x_j, \hat{y}) - \frac{1}{|C|} \sum_c a(n_i, x_j, c)$ where $\hat{y} = f_\theta(x_j)$ is the predicted class of $x_j$ and $a$ can be any of the attribtution function among [1; 4; 5; 6]. We choose to use DeepLIFT [5] score, and in experiments we find these attribution methods do not differ much.

**Spectral Co-clustering.** Given the constructed matrix, we use spectral co-clustering to find biclusters with values higher than those in the corresponding other rows and columns. Given a predefined number $k$ of biclusters, spectral coclustering algorithm treats the input data matrix as a bipartite graph and approximates the normalized cut of this graph to find heavy subgraphs. In the resulted biclusters, each row (neuron) and each column (sample) belongs to exactly one bicluster with no overlaps.

## 4 Experiments

We first inspect DNNs for visual classification on MNIST. We fetch a pre-trained model that is used and evaluated in the DeepLIFT paper [5]. The architecture is a 4 layer feedforward rectified model where the first two layers are convolutional layer followed by two fully-connected layer. (We also test on other DNN models with rectified neurons trained such as multilayer perceptron which has no convolutional layer, and the results are very similar.) We use 10000 data samples from the test set and choose $l$ to be the second convolutional layer of shape $(5, 5, 64)$. The total number of hidden neurons in $l$ is 1600. The choice of layer is rather arbitray, we choose it to be the second convolutional layer because we want the rows and columns to be roughly of the same magnitude, so that the neuron-sample matrix would have a good visualization, and that the biclustering algorithm can work properly. We compute the DeepLIFT score [5] and construct the neuron-sample matrix. We set $k = 10$ in spectral co-clustering as is the number of classes for MNIST. Different attribution

methods do not differ much in the qualitive observations, so we only present the comparision of different attribution methods on the ablation study in Figure 5b.

**The neuron-sample matrix is highly sparse.** The constructed neuron-sample matrix is very sparse, as only a few entries have attribution score above zero (see Figure 1a). Rearranging the display of the matrix by bicluster label, we can observe a checkerboard pattern in Figure 1b, indicating that certain groups of hidden neurons are indeed highly-correlated for predicting certain data samples.

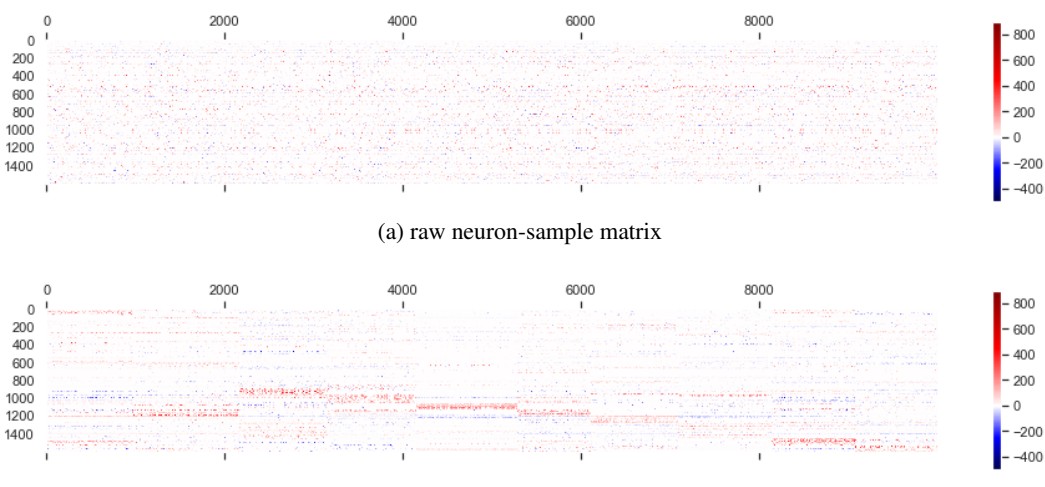

(a) raw neuron-sample matrix

(b) neuron-sample matrix rearranged with bicluster labels

Figure 1: *Visualization of the neuron-sample matrix* (Better see in digital version) Each row corresponds to one of the 1600 neurons in the second convolutional layer ($5 * 5 * 64$). Each column represents one of 10000 data samples.

We further report the distribution of attribution scores in Figures 2a and 2b, where the x-axis represents the attribute score and the y-axis represents the frequency. The overal distribution is centered around zero but inside one bicluster, the mass is concentrated on the positive side. This indicates that the spectral coclustering algorithm does find good subgraphs that a subset of hidden neurons typically have higher attribution scores on a subset of samples.

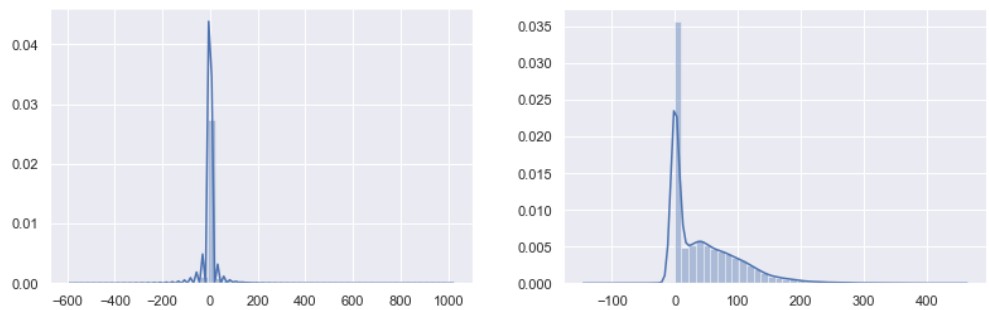

(a) overall distribution of attribution score in the raw neuron-sample matrix
(b) distribution of attribution score in the 0-th bicluster

Figure 2: *Distribution of attribution score.*

**Samples corresponding to the same bicluster show coherent feature patterns.** We find that the samples in one bicluster, i.e., samples corresponding to the same Functional Module , show coherent and similar feature patterns. We pick a bicluster and randomly gather samples from that bicluster and find these samples show very similar patterns (Figures 3a and 3b) and belong to the same class. We also show the distribution of true labels in the total $k = 10$ biclusters (Section 4). Note that despite

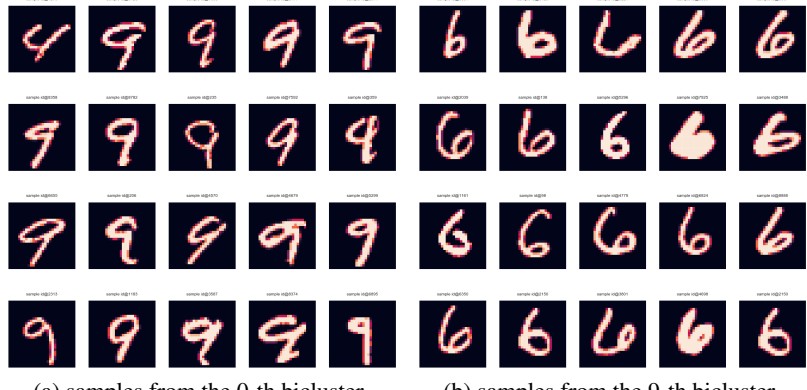

(a) samples from the 0-th bicluster      (b) samples from the 9-th bicluster

Figure 3: *samples from biclusters*

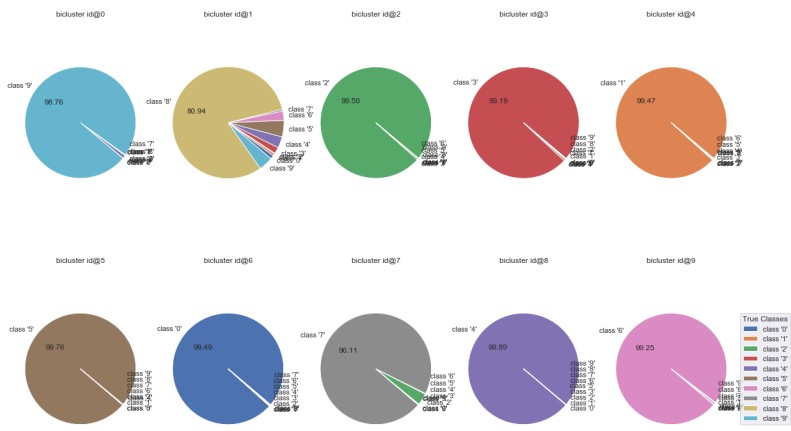

Figure 4: *Ground-truth label distribution in biclusters*

not using any label supervision in our approach, we can achieve a very good job of unsupervised discriminating.

**Spatial relationship of the neurons in a module: Is this modularity a result of convolution?.**
Surprisingly, the neurons in a bicluster do not seem to have any spatial relationships, such as being in the same channel or centered in some position. We also test on multilayer perceptrons where there is no convolutional layers and the above phenomenon can still be observed.

**Ablation Study: Functional Modules are critical for discriminating samples.** We check the performance of DNN by removing those hidden neurons by bicluster. In this experiment, we split the $10000$ images $X$ further into two sets where the first $5000$ samples as validation set are processed by our approach and the rest $5000$ samples are held-out for testing. We compare the accuracy when hidden neurons in layer $l$ are gradually ablated on the test set with several baselines for ablating neurons, such as ablation of random neurons, ablation of top-important neurons greedily, and ablation of neurons by module (bicluster) (ours). Ablation of a neuron is implemented by setting the activation value of this neuron to zero, as convention in previous studies [1; 6]. Note that we set $k$ larger in order to get a smoother curve. As shown in Figure 5a, ablation by module achieves the most significant accuracy drop compared with greedy ablation of top-k neurons (all use DeepLIFT score) and random ablation. It demonstrates that these Functional Module s found by spectral coclustering do play an important role in discriminating data samples.

We also compare the effect of abaltion with different attribution methods in Figure 5b and observe no significant difference among different choices of attributions methods.

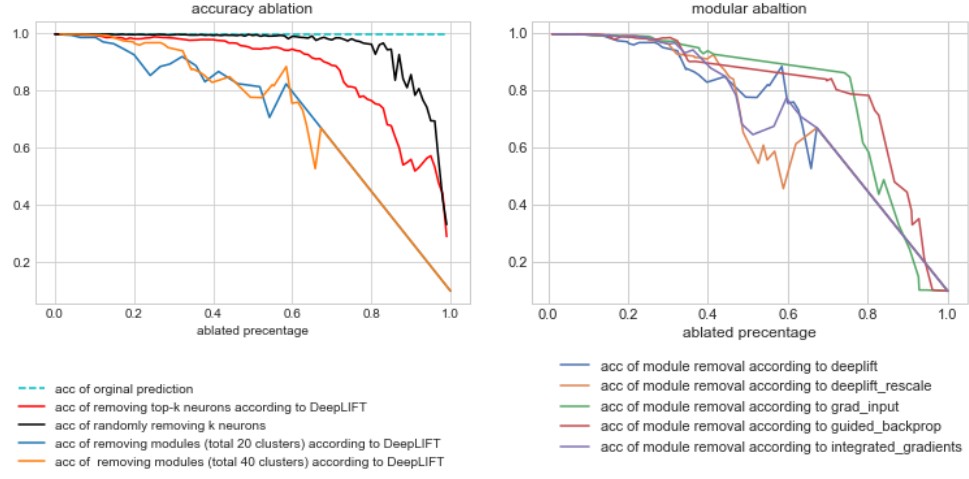

(a) ablated using DeepLIFT by different ways

(b) ablated with different attribution methods by modules (40 biclusters)

Figure 5: Accuracy decreases when neurons are ablated.

## 5 Conclusion

We develop an approach to parcellate a hidden layer into functionally related groups which we call Functional Modules , by applying spectral coclustering on the attribution scores of hidden neurons. We find the Functional Modules identifies functionally-related neurons in a layer and play an important role in discriminating data samples.

One major limitation of this short paper is that we have not tested on more general cases, such as different layers, different activation function, different models trained on more diverse datasets, etc. In order to gain generalizable insights, such a massive investigation is neccessary.

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
