# OpenReview forum: "Checking Functional Modularity in DNN By Biclustering Task-specific Hidden Neurons"
_NeurIPS.cc/2019/Workshop/Neuro_AI — Real Neurons & Hidden Units @ NeurIPS 2019 Poster_

### Official Review · AnonReviewer3 · 2019-09-20
**Novel technique for clustering neurons but not carefully evaluated or motivated.**

**Clarity:** 2

**Comment:**

A potentially interesting and important technique for clustering. More motivation for the decisions in their study of this algorithm. Figures need to be improved in terms of contrast and relative sizes of figure labels.

**Category:**

AI->Neuro

**Clarity Comment:**

Unclear whether units under study from MNIST were rectified or not. This would have a big difference on attribution depending on how sparse units were.
Figure 1a and c are not well scaled and the contrast is very low. A log scale might have helped and larger labels.
In line 65 it is not clear what 'the constructed matrix' is (neuron-sample, e, a).
Similarly figure legends/labels in Figure 2 C are overlapping and tiny. Not vectorized so pixellated.

**Evaluation:**

1: Very poor

**Importance:**

2: Marginally important

**Importance Comment:**

If they exist, finding functionally different groups of units in a DNN and using them to generate hypotheses in the brain is an important goal. The authors apply their technique to the second layer of a network trained to recognize digits. No evidence is provided that this network is functionally similar to the brain or that their technique would generalize to more complex ‘sensory networks’. Thus it is difficult to tell from the paper whether their potentially important technique is important.


**Intersection:**

2: Low

**Intersection Comment:**

They do mention that this method of clustering could be applied to neurons. Does not mention how this might work in practice or what would be gained or why it should be preferred over other clustering methods that have been applied to neurons.

**Rigor Comment:**

Many decisions were not well motivated, justified or described at least in brief. For example: choice of attribution method, choice of network and layer, number of clusters.  The space of parameters and network should have been explored more thoroughly to be convincing. Or at least justification for why their methods did not require more thorough testing.

**Technical Rigor:**

2: Marginally convincing

---

### Official Review · AnonReviewer1 · 2019-09-24
**interesting direction but more rigorous analyses are needed**

**Clarity:** 2

**Comment:**

It could be an interesting direction in terms of looking for functional modules in artificial neural networks via clustering methods but more rigorous analyses and clearer explanations are needed.

**Category:**

AI->Neuro

**Clarity Comment:**

All figures are too small to follow and understand, especially figure 2c which has labels overlapped. The grey tone color of neuron-sample matrix makes figure 1 hard to visualize.

**Evaluation:**

2: Poor

**Importance:**

2: Marginally important

**Importance Comment:**

It is potentially useful to show if functional modularity exists in artificial neural networks. The authors claim they find functional modules by applying biclustering to the neuron-sample matrix but current analyses primarily evaluated on MNIST dataset are not clear and limited to tell whether the existence of functional modules.

**Intersection:**

2: Low

**Intersection Comment:**

Although the authors claim the usage of neuron attribution and biclustering can be applied to real neurons, there is no explanation of how exactly to apply. Other than that, this paper is mainly focusing on artificial neurons.

**Rigor Comment:**

There is no explanation of the selection of tested layer and no comparison between results of other layers. Similarly, the choice of the number of biclusters is also confusing and no explanation is provided. In figure 2c, it looks like multiple biclusters contribute to the same class, then why not change the number of biclusters and evaluate more results.

**Technical Rigor:**

2: Marginally convincing

---

### Official Review · AnonReviewer2 · 2019-09-26
**Improving interpretability by analyzing bulk-behavior of subsets of neuron**

**Clarity:** 3

**Comment:**

Several questions are unanswered. The experiments demonstrate preliminary investigations, at best. More rigorous experimentation and crossvalidation are needed.
Is the phenomenon observed across different datasets with more diversity?
Is the possibility of memorization/overfitting ruled out? Is that even a concern, or is memorization what functional modularity implicitly refers to?
Does the fact that the network is convolutional help in identifying the biclusters because now the neurons are more structured than fully-connected networks?
The choice of attribution scores etc. is not clearly justified.

**Category:**

Neuro->AI

**Clarity Comment:**

Technical details were omitted due to space constraints. The main idea and outline of implementation and methodology were explained. Other works considering attribution and importance scores have been cited clearly.

**Evaluation:**

3: Good

**Importance:**

4: Very important

**Importance Comment:**

The technique could potentially improve the interpretability of DNN models for feature understanding

**Intersection:**

4: High

**Intersection Comment:**

A technique is discussed with application towards analyzing DNN models which are prevalent in machine learning/AI. This technique aims to make models and features learned by DNNs more interpretable, and the principle is inspired by biological hypotheses of functional modularity.

**Rigor Comment:**

Some rigor is needed in the experimental evaluation. The DeepBind study is not clearly explained and lacks technical rigor or has not been explained clearly. The claim "DeepBind model learns something from raw data" is too vague.

**Technical Rigor:**

3: Convincing

---

### Decision · Program_Chairs · 2019-10-02

Accept (Poster)